# A Buckycatcher in Solution—A Computational Perspective

**DOI:** 10.3390/molecules28062841

**Published:** 2023-03-21

**Authors:** Filipe Menezes, Grzegorz M. Popowicz

**Affiliations:** Institute of Structural Biology, Helmholtz Zentrum Muenchen, Ingolstädter Landstr. 1, 85764 Neuherberg, Germany

**Keywords:** semi-empirical, buckycatcher, π stacking, thermodynamics of binding

## Abstract

In this work, we study the buckycatcher (C_60_H_28_) in solution using quantum chemical models. We investigate the conformational equilibria in several media and the effects that molecules of solvent might have in interconversion barriers between the different conformers. These are studied in a hypothetical gas phase, in the dielectric of a solvent, as well as with hybrid solvation. In the latter case, due to a disruption of π-stacking interactions, the transition states are destabilized. We also evaluate the complexation of the buckycatcher with solvent-like molecules. In most cases studied, there should be no adducts formed because the enthalpy driving force cannot overcome entropic penalties.

## 1. Introduction

The subject of non-covalent interactions is a fascinating field of primary importance to most of the research fields related to chemistry [1]. From interactions between drugs and receptors in medicinal chemistry and pharmaceutical sciences [2] to the behavior of condensed phases in thermodynamics [3,4], the interplay between atoms that are not bonded to one another determines many of the micro- and macroscopical properties of systems.

Non-bonded interactions play a particularly relevant role in the field of supramolecular chemistry, which attempts to use specifically these forces to design complex and organized structures from simpler molecular units [5,6,7], much as biological systems are made. Ever since their discovery by Sygula et al. [8], buckybowls have attracted the attention of the scientific community. Due to a careful molecular design, this derivative of corannulene has the ideal curvature to capture fullerenes C_60_ and C_70_ [8,9,10,11,12,13,14,15,16,17]. This curvature is key for binding, since it promotes strong ball–socket π interactions that keep the complex together [8].

From the theoretical point of view, modelling the adducts of the corannulene pincer and fullerenes is complex. This has mostly to do with the fact that many Density Functional Theory (DFT) methods are unable to correctly describe the non-bonded interactions in such structures. Π stacking arises from the interplay of dispersive forces and Pauli repulsion between the electronic clouds of π-systems [18]. The (mostly) attractive contribution from dispersion, is however not well covered by lower-level quantum chemical methods, such as DFT [19,20] or it is even completely absent, as in Hartree–Fock (HF) [19]. This is, furthermore, passed over to simplified (semi-empirical) variants of these methods.

In principle, ab initio methods should properly describe π-stacking interactions; however, one requires relatively high-levels of theory. MP2′s inability to describe dispersion interactions associated with π systems is no exception in this case [21]. With a predicted binding energy of 78.3 kcal/mol [16], the deviation of MP2 results towards the latest reference (experimentally back-corrected) energies is about 50 kcal/mol (28.4 ± 0.6 kcal/mol) [22]. In principle, quantum chemistry’s holy grail Singles Doubles Coupled Cluster with Perturbative Treatment of Triples (CCSD(T)) should correctly describe the binding energies of such structures. The very steep scaling of canonical CCSD(T)’s computational cost with the molecular size (O(N^7^)) makes it impracticable to use on systems of the size of C_60_, let alone a complex of this fullerene with the corannulene pincer.

From the perspective of quantum chemical simulations, the situation improved significantly with the introduction of force-field-like corrections that account for dispersion [19,20,23,24,25,26,27,28,29,30,31,32,33,34,35,36]. With these, even lower-level semiempirical methods can accurately predict the binding energy of buckycatcher-fullerene complexes [37,38].

Despite these achievements, many questions about the corannulene pincers and its complexes with fullerenes remain open. Binding free energies have been measured experimentally, and the change in entropy upon binding was measured to be slightly positive (∼0.5 cal/[K.mol]) [11]. Theoretical estimates are all negative [39]. There is also a great discrepancy between gas-phase enthalpies and the experimental values. Interactions with the solvent have been brought to the discussion [11]; however, since much is unknown about the behavior of the buckycatcher in solution, it all remains speculative. Solvent molecules have, however, been found trapped in crystals of the buckycatcher [9]. Careful analysis of the crystal structures shows a tendency for the buckycatcher to encapsulate what is available.

Inspired by the observation of Zabula et al. on “what would buckycatcher catch if there is nothing to catch” [10] and the observations regarding the role of the solvent in the formation of adducts between the corannulene pincer and fullerenes [11], we decided to employ our newly developed semiempirical package to tackle some of the questions still associated with the buckycatcher system (Figure 1).

Using several dispersion-corrected semi-empirical methods and solvation models, we explore the conformational equilibria of the buckycatcher and its explicit interactions with typical solvents. CCSD(T) and DFT calculations are used to infer on the accuracy of the semi-empirical predictions. Our hope is to shed further light on the role potentially played by the solvent in the formation of adducts between the buckycatcher and fullerene, which has been disregarded in the literature.

## 2. Results

### 2.1. Systems Studied and Nomenclature

As established in the literature [13], the corannulene pincer C_60_H_28_ is described by four main conformations. The most known of these is the structure in which the two corannulene units point inwards to form a convex shape (*ii* for in–in). Other open conformations may be found by giving the pincers a concave shape (*ee* for ex–ex) or to round the arms of the pincers in the same direction (*ie*). The latter conformer may exist furthermore in a tight/closed form (*iet*).

This work considers the formation of adducts with small, solvent-like molecules. These include toluene, acetonitrile, n-hexane, chloroform, dichloromethane and two conformations of 1,1-2,2-tetrachloroethane. The latter, we denote as *anti* if the two chlorine atoms on each carbon point in different directions and *asym*(metric) if otherwise. In the case of toluene, we consider adducts involving two and three molecules of solvent. Information on the adducts with solvents as well as more detailed information on the respective point group symmetries may be found in the Appendix A.

It is important that we define binding energy as the negative of the energy for the aggregation reaction. Taking, as example, the general reaction A + B ⇌ AB, the reaction’s energy is defined as E_rx_ = E_AB_ – E_A_ – E_B_, and the binding energy E_bind_ is the negative of E_rx_. Consequently, a negative binding energy describes an unstable complex.

All structures we optimized will be available in a gitlab repository under the link https://gitlab.com/siriius/buckycatcherinsolution.git (accessed on 18 February 2023) upon publication of this manuscript. A comparison with some experimental crystal structures is provided in the Appendix A.

### 2.2. Conformations of the Corannulene Pincer

All conformers of the corannulene pincer were readily optimized at all levels of theory considered, except for *iet*, which does not exist on the PM6 Potential Energy Surface (PES). This is because the method has a poor description of London forces via a Lennard—Jones-like contribution in the nuclear repulsion term. Attempts to optimize *iet* always resulted in *ie* instead. A simple means of characterizing the pincers and how widely open the arms are is via the distance between the two outermost carbons of the corannulene units located along the plane of symmetry conserved in all species (R_P_ in Figure 2) as defined by Zhao and Truhlar [12].

We observed that dispersion-corrected PM6 methods yield identical descriptions for all conformers (*c.f.* Table 1). Uncorrected PM6 leads to similar results for open structures, for which, we conclude that, expectedly, the relative stability of these species is not dominated by dispersion forces. We calculated the contributions that would arise from three-body dispersion, and these played no role whatsoever on the relative energies of the conformers. Open conformers at the GFN2-xTB level show wider openings in comparison to PM6 methods (larger R_P_); however, π-stacking forces in *iet* are stronger and hold the pincer’s arms closer together. Our R_P_ values match reasonably well with the literature data [12,13]—in particular, the GFN2-xTB calculations.

Irrespective of the method, the difference in energy between open conformers is always within 1 kcal/mol (Figure 3 and Table 2). Except for the GFN2-xTB results, *ii* is systematically predicted to be the most stable open conformer of the buckycatcher. On the other hand, *ee* is expected to be the least stable open conformation. The GFN2-xTB results expect the opposite trend, although, in this case, we consider all results within the accuracy of the semi-empirical methods employed. This further matches the DFT results of others [8]. We consider, therefore, for practical purposes, the three open forms of the corannulene pincer to be isoenergetic. Unless there are kinetic or entropic impediments at interplay, these conformers should all contribute almost equally to the ideal gas-phase compositions.

Interestingly, discrepancies arise in the relative stability of *iet* with respect to the open conformers: GFN2-xTB and the ab initio data (coupled cluster calculations) predict *iet* to be the most stable form for the corannulene pincer by at least 4 kcal/mol with respect to other conformers; PM6-based methods and M06-2X penalize the formation of this species. We note that such disagreement has been observed with previous DFT calculations: the B97-D/cc-pVQZ calculations of Mück-Lichtenfeld et al. [13] expect *iet* to be the most stable species.

Denis and Iribarne [17] made similar observations at the B97 level using other basis sets as well as with the B3LYP functional. Denis and Iribarne [17] observed that, with the M06-2X and PBE functionals, *iet* is expected to be the least favorable conformer. We believe, however, that our Coupled Cluster calculations set clear what is the order of stability for the different conformers, in this case, with GFN2-xTB as the overall most accurate semi-empirical method (the energy differences between open conformers are very small and can well be fortuitous in the case of PM6 calculations).

### 2.3. Equilibria between Conformers

Irrespective of the method, the equilibrium between open conformations is not dominated by entropy since the slopes of the Gibbs free energies with temperature are all negligible (Figure 4). Differences in energy are also small, particularly with PM6-D3H+. Ideal gas thermodynamic data further strengthens our previous conclusion on the relative weight of the three open conformations for the Boltzmann statistics. In the case of GFN2-xTB, there is a larger entropy penalty over *ii*. Though barely visible in the plot of entropy against temperature, such effects accumulate to make *ii* the least stable species in gas. The results for PM6-D3H4X are qualitatively equivalent to the data shown for PM6-D3H+, with different enthalpy values for conformer *iet*. This is available in the Appendix A.

When bringing the equilibria to the dielectric medium of toluene, the ALPB model differs very little from the respective gas-phase data (c.f. row $\Delta G_{PhMe}^{GFN2/ALPB}$ in Table 3). The equilibrium most affected is the one between *ie* and *iet*, mostly due to differences in solvent accessible surface areas. Changes are, nevertheless, below 0.5 kcal/mol for all cases. Although we used COSMO only as parametrized for PM6-D3H4X, we mixed these solvation energies with other models to best evaluate the effects of the solvation model. Qualitatively speaking, the relative stability of open conformations remains unchanged. However, at the GFN2-xTB level, COSMO penalizes conformer *iet*, which reaches a chemical potential identical to *ee* and *ie*.

At the PM6-D3H4X and PM6-D3H+ levels, the conclusions are opposite: *ii* is expected to be the most stable species; the additional penalty from COSMO makes *iet* completely irrelevant in solution (at most, a concentration of 0.1%).

In Table 3, we have the ALPB results for the conformational equilibria in other solvents. This includes data for dichloromethane, chloroform, benzene, phenol and acetonitrile. Despite differences in the nature of the solvents, the results are very uniform: there is a minimal shift in the relative stability of open conformations in going from the gas phase to the different media; *iet* is always the least stable species. In fact, *iet* becomes at least twice as unstable compared to *ii*, which is penalized at the GFN2-xTB level. We conclude that toluene as a solvent is an outlier in the ALPB model, and the results appear to be untrustworthy. Based on this analysis, we decided to use, from here on, benzene as a replacement solvation model of toluene at the ALPB level. This means that, when discussing toluene’s solvation by ALPB, we mean that benzene’s parametrization was used.

### 2.4. Interconversion between Conformers

To better understand the behavior of the corannulene pincer, we optimized the transition states bridging the different open conformations of the buckycatcher. In any of the cases, the saddle points have the respective corannulene units as completely flat. Figure 5 schematically presents the energy landscape connecting the three open conformers of the corannulene pincer according to the GFN2-xTB and dispersion-corrected PM6 methods. Irrespective of the method chosen, calculated barriers for the two transitions are very similar, which match quite well the 11 [13] and 8 kcal/mol [17] reported in the literature. PM6-based activation barriers are quite accurate in this case, which we justify with the fact that there is no bond-breaking or forming in the transitions studied.

The main difference between the semi-empirical calculations is the relative order of the heights of the barriers, which result primarily from the relative energies of each conformer. Here, PM6-D3H4X and GFN2-xTB are the semi-empirical methods closest to the high-level Coupled Cluster results (Table 4). The very fine details leading to the relative heights of the barriers are better captured by PM6-D3H4X. Irrespective of the set of geometries taken, the barrier to form the most extended conformer is expected to be the highest. Solvation calculations using the ALPB model show a decrease by less than 0.2 kcal/mol in the activation barriers when the toluene’s dielectric is used.

### 2.5. Explicit Interaction with Toluene

We identified two sets of complexes between toluene and the corannulene pincer. These are described by different relative orientations of toluene in the cage formed by the catcher. The first set of complexes has the molecule of solvent quasi parallel to both corannulene units, and it is close to perpendicularity with respect to the tethers. The second set of complexes has toluene parallel to one arm and perpendicular to the other. Due to the nature of interactions, the former set shows lower R_P_ values than does the latter.

Interestingly, we were thus far unable to locate the second set of conformers using the ALPB/toluene solvation model. We always required an increase in the polarity of the medium to stabilize the species and obtain a stationary point (for instance, in acetonitrile). At the PM6-D3H4X/COSMO level, all structures were readily optimized with any of the dielectric constants. Despite the different binding pose, the binding affinity was similar in both sets of conformers. We, therefore, relegated the data on the second set of complexes to the Appendix A. Data on the first set are available in Table 5.

Although the energetics are favorable for binding, the entropy penalties for the formation of adducts are always large and similar in magnitude. It is particularly interesting to note that the calculated entropies are very similar even between the different semi-empirical methods. This means that, from the thermodynamic point of view, the calculated vibrational frequencies are of equivalent accuracy. What distinguishes methods are the binding energies (enthalpies)—namely, the binding involving conformers *ii* and *ee*. It is important to stress here that all entropies we calculated correspond to gas-phase corrections. Entropic contributions, such as those of the explicit solvent, are disregarded.

When enthalpies and entropies are weighted together at the GFN2-xTB level, only conformer *ii* could potentially form stable adducts with toluene in the gas phase. The case of the other two conformers is borderline, since the absolute values of the respective Gibbs free energies for forming the adducts are below 1 kcal/mol. For the other methods, the Gibbs free energies for the formation of adducts are clearly positive, meaning that these aggregates are not expected to spontaneously form in the gas phase.

Solvation always contributes to the destabilization of complexes, and the effect is so large that, in no case, is complex formation in solution to be expected. Though qualitatively COSMO and ALPB agree, the magnitude of solvation effects is again quite different. For a more accurate evaluation, conformer statistics should be included for a single Gibbs free energy value. However, as binding is clearly not expected to be spontaneous, we did not perform such calculations.

We explored the possibility of forming higher complexes of the buckycatcher with toluene. The results are, however, identical: though favorable from the enthalpy point of view, the entropy penalties were consistently too large. Not even in the gas phase were the Gibbs free energies for forming the adducts positive. All the respective thermodynamic data are provided in the Appendix A.

To double check the results of the semi-empirical calculations, we took the GFN2-xTB-optimized complexes, and we reevaluated energies at the DLPNO-CCSD and DLPNO-CCSD(T) levels. The binding of toluene to the buckycatcher became stronger by, respectively, 4.6 and 7.8 kcal/mol, with respect to the GFN2-xTB data. Irrespective of the high-level method, the difference in energy is so significant that, when we include the ALPB solvent correction, stable adducts are expected to be formed.

This is so astonishingly large that, for the time being, we refrain ourselves from further discussion. This is relegated for a later stage of the present manuscript after presenting more results. Thus, neglecting for the time being, the high-level calculations, the semi-empirical results indicate that, although toluene may transiently stay within the pincer’s arms, the formation of a stable adduct is not to be expected. This result is consistent for all semi-empirical methods used, despite some quantitative differences between the different models.

We also studied the explicit inclusion of toluene on the interconversion barriers between the open conformers of C_60_H_28_. Comparing with the free case, the presence of toluene induces the pincers to come together for the transition between *ie* and *ii* (R_P_ of 8.09 Å instead of 12.88), which forces the transitory arm to bend slightly inwards. In the case of the transition between *ee* and *ie*, the presence of toluene changes the R_P_ by less than 0.05 Å.

Though the flattening of the pincer’s arms could potentially be favorable to interact with toluene’s aromatic ring, the methyl group weakens the possible stabilization of the transition state via π stacking. Toluene in between the pincer’s arms, thus, increases the GFN2-xTB activation barriers to 13.4 (*ii* → *ie*) and 11.2 (*ie* → *ee*) kcal/mol. Note that the second barrier is not affected as much as the first one. This is because of the additional bending that toluene induces on the pincer’s arms for the transition of *ie* to *ii*.

The ALPB model has a lowering effect in both transition states. For the formation of *ie* from *ii*, solvation lowers the Gibbs free energy by 2.3 kcal/mol. For the other transition, the effect is larger, with a lowering of 3.4 kcal/mol. The net effect is, thus, an increase (by 2.3 kcal/mol) of the activation barrier for converting *ii* to *ie* and a decrease by 0.8 kcal/mol for the conversion of *ie* to *ee*. In solution, one expects toluene between the pincer’s arms; thus, it is possible that there will be a split in the rates of interconversion between the different conformers.

### 2.6. Binding to Tetrachloroethane

Table 6 condenses the main thermodynamic data at 300 K for the specific binding of tetrachloroethane A (TCA), tetrachloroethane B (TCB) and of tetrachloroethane as a mixture of conformers to the buckycatcher. Note that, in all cases, conformational equilibria of the catcher is included, and, in the last case, we considered the conformational equilibria of all species.

The binding enthalpies at the PM6-D3H+ level are more attractive by 5-6.5 kcal/mol than at PM6-D3H4X or at GFN2-xTB. This is a consequence of attractive two-body dispersion. For instance, the R_P_ value for the complex TCA@ii optimized at the PM6-D3H4X level is 7.50 Å, whereas, at PM6-D3H+, it takes the value of 6.85 Å. As PM6-D3H4X and GFN2-xTB are two completely disparate methods and yet they give identical results, we tend to trust these more in detriment to the PM6-D3H+ data.

Based on the construction of the methods we analyzed, we conclude that PM6-D3H+ would benefit from the inclusion of three-body dispersion [22]. On the other hand, based on the similarity of the PM6-D3H4X and GFN2-xTB results, it is not straightforward to conclude the same for PM6-D3H4X. Entropies of binding are, in all cases, rather similar and large.

Gas-phase Gibbs free energies are, in all cases, negative. We note, furthermore, that the total binding of tetrachloroethane is more favorable than the specific binding of TCA or TCB. This is due to extra stabilization from the conformational entropy of the aggregated state, which helps to shift the equilibrium towards the formation of adducts. Introduction of solvation effects always raises Gibbs free energies with respect to the gas phases. Though severely hindered by solvation, the complexation of tetrachloroethane is expected to be thermodynamically favorable.

### 2.7. Caging of Other Solvents

In this section, we study the binding of other solvents to the buckycatcher. Results are shown only for GFN2-xTB. Furthermore, binding is studied for conformer *ii* of the catcher. In the Appendix A, we provide the respective thermodynamic data for other open conformers of the pincer. The main differences are in the enthalpies of binding, which are lower for *ie* and *ee*. Table 7 condenses the main thermodynamic data.

Compared to free C_60_H_28_, the presence of a single molecule of toluene will promote the closing of the pincer’s arms, forming a van der Waals cage for the solvent. The value of R_P_ when toluene is inside the pincer is 6.61 Å, which should be compared against the 12.01 Å for the free catcher in vacuum. A similar behavior was observed for all other solvents studied, where, however, R_P_ varied mainly according to the volume of the caged molecule. We obtained R_P_ values of 6.68 Å for acetonitrile, 6.84 Å for dichloromethane, 7.21 Å for chloroform, 7.32/9.31 for tetrachloroethane (respectively for TCA and TCB) and 8.01 Å for n-hexane.

We find that binding is always energetically favorable for the cases we considered, even for a molecule, such as n-hexane. This agrees with the values of R_P_ reported above. The key question is then not whether binding is possible but rather how strong the enthalpy is compared to the entropic penalty for forming the adduct.

To the good extent one observes a linear correlation between the enthalpies and entropies of binding, these are, however, split into two groups according to the size of the solvent (c.f. Appendix A). For small solvent molecules (i.e., excluding n-hexane and toluene), we found that an enthalpy of binding of about −9 kcal/mol would yield a gas-phase Gibbs free energy of 0 kcal/mol. Of the possible gas-phase complexes, only n-hexane showed (slightly) positive Gibbs free energies of aggregation at 300 K. Though favorable, acetonitrile’s case can well be within the limits of accuracy of GFN2-xTB. All other solvent cages in the gas phase are expected to be thermodynamically favorable.

The addition of solvent dielectric effects contributes in all cases to the increase of Gibbs free energies of complexation. No binding is expected with the buckycatcher at room temperature. High-level binding energies were also obtained for the complexes with acetonitrile and chloroform. Table 8 shows the respective results compared against GFN2-xTB.

Similar to the case of toluene, the CCSD and CCSD(T) binding energies are significantly shifted towards the formation of the complexes. This is particularly troublesome in the case of the complex with acetonitrile, where the differences amount to over 5 kcal/mol. Nonetheless, no conclusion whatsoever is altered by the refined dataset. For chloroform, the difference of 3.8 kcal/mol implies that a complex in solution is expected to be stabilized.

## 3. Discussion

Of the three open conformations of the corannulene pincer, one acts as an efficient van der Waals trap for other molecules. This is conformer *ii*, the one typically assigned to the catcher. Though the other two open forms can still establish reasonably strong interactions with small molecules, the curvature of their arms is inadequate for efficient trapping. This is reflected in the larger enthalpies of binding with small molecules.

When the buckycatcher is in a vacuum it will have, from the enthalpy point of view, the tendency to catch any molecule it finds. Even if this molecule can only participate in weak attractive interactions, such as n-hexane, there is at least one clear minimum along the interaction surface. Entropy, however, empowers the buckycatcher with selectivity, and one is to expect the preference to capture molecules that can yield sufficiently strong dispersion interactions.

The same applies to other conformers, although the enthalpies of binding are typically lower. At sufficiently low temperature and in the absence of an adequate partner (or the absence of a partner altogether), the buckycatcher should prefer to “catch itself” and collapse to form predominantly *iet*. Note that this might require crossing at least one barrier, which, in the literature, is estimated to lie at about 7 kcal/mol [13]. Other barriers that might be involved are the interconversion between different conformations.

For PM6-based methods *ii*, *ie* and *ee* are all equivalent from the energy point of view. GFN2-xTB disfavors *ii*. Strictly looking at relative electronic energies, *ii*, *ie* and *ee* should all contribute equally to the Boltzmann statistics of gas phases (also at the GFN2-xTB level). This is corroborated by higher-level calculations. When entropy effects are considered, chemical potentials at the GFN2-xTB level show an asymmetry in the contribution of the different species, which is transmitted to condensed phases. The composition of *ii* is expected to lie at 20% instead of approximately 33%. On the other hand, *iet* is expected to be the most stable conformer of the buckycatcher. This was only recovered by GFN2-xTB.

Though the barriers of interconversion between the different conformers are not large, we note that, in a toluene environment, one is to expect an increase in the activation barrier for forming *ii* simply because toluene in between the pincer’s arms destabilizes transition states with respect to reagents or products. Though we did not study other solvents, we expect that the global effect of solvation on the transition barriers should be greatly solvent dependent, and a hybrid solvation model with explicit molecules should yield more realistic results. For instance, we would expect chlorinated solvents to affect the interconversion barriers in a similar fashion to toluene. On the other hand, other solvents studied might not exhibit any significant effect, such as n-hexane.

Another interesting observation concerns the effects of solvation on the stability of adducts with solvents. Of the molecules tested, only 1,1-2,2-tetrachloroethane could potentially form clearly stable adducts with the buckycatcher in solution. Binding is particularly favorable between conformer *ii* and TCA. These two species together should promote and drive the binding process of tetrachloroethane. The respective adduct is expected to be the main constituent of the equilibrated solution.

In all other solvent case-studies, the enthalpy stabilization is insufficient to counter- act the entropy penalties. It is interesting to note the effects that contribute to the stability of the complexes. On one side, increasing the number of chlorine atoms leads to stronger interactions, thus, potentially favoring binding. This would be the case of dichloromethane against tetrachloroethane. On the other hand, too many chlorine atoms on the same carbon weakens binding: dichloromethane against chloroform. This could be related to binding deformation energy contributions. The other case studies that we presented further stress the preferences of the buckycatcher: molecules that allow strong dispersion-like interactions. Note that toluene is adequate for π stacking, however, the enthalpy of interaction is too weak for efficiently caging the molecule (according to semi-empirical levels).

At this point, we need to address the differences between semi-empirical methods and the higher-level DLPNO-CCSD and DLPNO-CCSD(T) for catcher–solvent complexes. We start by noting that there is a significant discrepancy between the CCSD and CCSD(T) results. Interestingly, this also happens for the relative stability of *iet* with respect to open conformers. Note that the former exists due to *π* dispersion, whereas the relative energies of the latter are almost independent of London forces. Given the way the ab initio methods are constructed, CCSD(T) should yield more realistic many-body dispersion compared with CCSD.

Nevertheless, we feel that the calculated binding energies are too strong. Irrespective of the magnitude of interaction energies, it is safe to conclude that many-body dispersion effects are of relevance for the systems here studied—an expected conclusion given the nature of the systems. Recent studies on DLPNO-CCSD(T)’s accuracy show that errors with respect to canonical CCSD(T) should grow linearly with the system size [40]. Judging from the results presented in the literature, for the thresholds we used, the basis set size and the systems studied, the errors could lie at about 2.0–3.0 kcal/mol.

In this sense, if the DLPNO-CCSD(T) binding energies are 2.5 kcal/mol too strong, the GFN2-xTB binding energy for CHCl_3_@C_60_H_28_ is within 1 kcal/mol of the higher-level ones. Still, the deviations between methods in the binding of acetonitrile and in toluene are too large. Contrary to the case of chloroform, these last two systems are dominated by *π* dispersion. By construction, DLPNO methods use MP2 for the so-called weak pairs. Weak pairs consist of contributions from doubly excited configurations, where the orbitals involved in the excitation are reasonably distant from one another. Each weak pair carries significantly less correlation energy than any strong pair, which builds the rationale for treating the former at a lower level of theory.

Nevertheless, because the number of distant pairs grows faster with the system size than does the number of close pairs, the total amount of correlation energy resulting from all weak pairs might become significant for large systems. This should be particularly true for the molecules that we are considering here. Irrespective of their nature (weak or strong), the pair correlation energy obtained by MP2 for *π* dispersion is known to be significantly overestimated.

It is then logical to ask ourselves whether the ab initio results that we report here are not being negatively impacted by using MP2 to calculate the contribution of weak pairs for the correlation energy. Though a deep enough analysis of our hypothesis goes beyond the scope of the present work, we believe that this could be a reasonable explanation for our report as well as for certain observations made in the literature [40]. Please note that we are not claiming any superiority for semi-empirical methods over ab initio theory. We are simply placing all data in perspective and giving more weight to the estimates for the binding energies of similar complexes [22]. Note that the diffusion Monte Carlo [41] and back-corrected experimental data [22] are consistent with one another. Both sets of results place the binding energies significantly below those calculated with DLPNO-CCSD(T).

We can, however, attempt to estimate the reasonability of our proposal based on the data that are currently available. For this, we used the results of our previous GFN2-xTB calculations [38], the CCSD(T) calculations of Villot and co-workers [42] and the results of the present document to recalculate the Gibbs free energies for the formation of adducts between the buckycatcher and the fullerene C_60_. If we assume a direct aggregation reaction, then we correct our previous GFN2-xTB results by −1.5 kcal/mol, resulting in the value of −14.4 kcal/mol.

Considering an exchange reaction between toluene and fullerene (PhMe@catcher + C_60_ D PhMe + C_60_@catcher), the Gibbs free energy increases to −9.6 kcal/mol. Both values are significantly off from the −4.8 kcal/mol obtained experimentally [11]. From those 2 values, one could propose that more molecules of toluene might be involved in the process. Even if the resulting Gibbs energies match the experimental data, enthalpies and entropies will not. The experimental change in entropy is very close to zero [11], which indicates a 1:1 exchange reaction.

In this scenario, we cannot invoke significant errors in the calculated entropies of binding, since our previous estimates [38] clearly indicated that, for 1:1 exchange processes, this contribution is, expectedly, very close to zero and to the experimental data. Errors must consequently be attributed to the enthalpy terms, and three main sources of error can be identified: electronic energies; vibrational, rotational and translation terms to enthalpy (RRHO from rigid rotor and harmonic oscillator); and solvation effects.

We begin by analyzing the solvent effects. Of all models that we previously tested [38], ALPB was the most penalizing one. Irrespective of its accuracy and precision, ALPB’s errors are, thus, potentially contributing to error cancellation, rendering the calculated process less spontaneous and bringing it closer to the experimental values. We note however that, in all studies, there is exclusive recurse to implicit solvation. It would be interesting to test these systems using explicit solvation and sampling methods. Next, we have the RRHO terms.

According to the discussion above, entropies are reasonably accurate; thus, large errors are not expected. This is further corroborated by calculations reported in the literature [22], where semi-empirical RRHO contributions to the Gibbs free energies are compared against those of DFT calculations: the results are of similar accuracy. We are then left with contributions from the electronic energies. Zhao and Truhlar [12] studied the formation of adducts between corannulene and C_60_. The advantage of this system is that it is very similar to the present case studies but also there is experimental data available: corannulene:C_60_ adducts do not form spontaneously in the gas phase [43].

M06-2X calculations of Zhao and Truhlar corroborate the experimental observations. The binding energy they calculated is around 12.4 kcal/mol. We calculated the binding energies using GFN2-xTB, and we obtained the value of 18.5 kcal/mol. Using the RRHO correction terms of Zhao and Truhlar, obtained at a DFT level, the increase in binding energy would imply a gas-phase Gibbs free energy of −1.3 kcal/mol for forming the adduct. We stress that the estimated spontaneity of the process goes against the experimental observations.

Though we did not calculate the binding energy for the corannule:C_60_ aggregate at the DLPNO-CCSD(T) level, all the data herein discussed shows systematically that binding energies calculated with GFN2-xTB are lower than those at the DLPNO-CCSD(T) level. Thus, we expect that using the linear scaling variant of CCSD(T) would make the Gibbs free energies for forming the corannulene:C_60_ adducts even more negative. Based on this analysis, it seems plausible that the DLPNO-CCSD(T) binding energies significantly contribute to the errors.

Returning now to the semi-empirical data, where only one solvent was found that could be captured by the buckycatcher, we raise the question of what is the actual role of the solvent when the buckycatcher captures fullerenes: is it simply in offering an environment that hinders binding, or is there an explicit participation in the thermodynamics of binding? It seems plausible that, in tetrachloroethane, there is an explicit participation of the solvent in the reaction, which is supported by the experimental data. However, the same does not seem to apply for toluene, which seems contradictory [11,38]. We stress that, in this case, GFN2-xTB/ALB would have to be incorrect by at least 4 kcal/mol in order to yield undoubtedly a spontaneous aggregation process that would then compete against encapsulation of a fullerene.

## 4. Materials and Methods

Calculations were performed using our newly developed library, ULYSSES [44]. Geometries were minimized using the Broyden–Fletcher–Goldfarb–Shanno (BFGS) algorithm with the dogleg trust-region method with strict convergence criteria—namely, 10^−8^ E_h_ for energies and 2.5 × 10^−5^ E_h_/a_0_ for gradients. The Hessian was approximated using the method of Lindh et al. [45]. Transition states were optimized using Baker’s Rational Function Optimization (RFO) [46] with convergence criteria of 10^−5^ E_h_ for energies and 5.0 × 10^−4^ E_h_/a_0_ for gradients. The geometry optimization of transition states used the numerical Hessian instead of Lindh’s.

The Hamiltonian of choice for optimizing geometries is always consistent with the method chosen for energy and Hessian evaluation. These are GFN2-xTB [37], PM6 [47], PM6-D3H4X [28,48,49] and PM6-D3H+ [50,51]. We wish to stress that, in the D3H4X correction, there are hydrogen repulsion contributions, which are present in the simulations involving the catcher. Thus, D3H4X accounts for more than just attractive dispersion effects. For D3H+, there is only the contribution from Grimme’s D3 correction, meaning that PM6-D3H+ is equivalent to PM6-D3.

For the calculation of thermodynamic properties, we used two models implemented in ULYSSES, which differ only in the vibrational partition functions. The first model is the traditional harmonic oscillator approximation as described in any textbook in statistical mechanics [52,53]. This model, however, is expected to be unreasonable for low-frequency internal modes, such as those characteristic of non-bonded aggregates. Consequently, we use the free-rotor/harmonic oscillator interpolation method of Grimme [39], which we ex- tended for thermodynamic quantities other than entropies [44].

Gibbs free energies of solvation at the GFN2-xTB level were estimated using the ALPB solvation model as described by Ehlert et al. [54] and as implemented in ULYSSES. ALPB is a regularized and approximately summed analytical expression for the linearized Poisson–Boltzmann equation. Despite its similarities with traditional Generalized Born theory, ALPB includes, for instance, size and shape considerations [55].

In some cases, we also calculated solvation effects using COSMO [56] as available from MOPAC [57] together with PM6-D3H4X. The solvent dielectric constant used for toluene solutions was 7.0, whereas, for the case of tetrachloroethane, we used 8.42. The former is higher than the experimental dielectric constant of toluene but in agreement with the parametrization of ALPB. We also calculated the COSMO solvation energies using the actual dielectric constant of toluene to verify there was no significant effect on the results.

The only solvent for which there was no parametrization available in the ALPB model was 1,1-2,2-tetrachloroethane. As the dielectric constant and the refractive index of dichloromethane are relatively close to those for tetrachloroethane, for simulations in the latter, we used the parametrization for dichloromethane. Furthermore, in the beginning of this work, we always used toluene’s parametrization in the ALPB model. We verified, however, that this parametrization leads to incorrect results, and thus we used benzene as a model replacement. Such a replacement is duly declared in the main text. To ensure that this problem was not related to our implementation, we verified that our calculations were numerically equivalent to the results from xTB [58].

In certain rare occasions, we were unable to obtain adduct geometries without imaginary vibrational frequencies, which are very hard to remove for flexible systems. In such situations, we used the absolute value of the respective imaginary vibrational frequencies in the calculation of thermodynamic properties as suggested by Sure and Grimme [22].

This simplification was, however, only employed with less relevant species, i.e., the ones for which an inaccuracy in vibrational modes would not possibly influence their thermodynamic relevance. Furthermore, in no case did we accept a structure with more than one imaginary frequency larger than 20i cm^−1^. Absolutely no vibrational frequency was disregarded in any part of this work. This speaks for the unprecedented high accuracy of the structures we are working with.

Plots were produced with Python’s matplotlib [59]. Figures of the molecules were generated with UCSF Chimera, as developed by the Resource for Biocomputing, Visualization and Informatics at the University of California, San Francisco, with support from NIH P41-GM103311 [60].

DLPNO-CCSD, DLPNO-CCSD(T) [61] and M06-2X [62] calculations were run on ORCA 5.0 [63,64,65]. Ab initio and DFT calculations made use of the def2-TZVP [66] basis set and resolution of the identity [67,68].

## 5. Conclusions

In this contribution, we analyzed the conformational equilibria of the corannulene pincer in gas phase and in several media using the COSMO and ALPB solvation models. We verified that, although gas-phase compositions are method-dependent, solution-phase compositions are somewhat consistent between GFN2-xTB and dispersion-corrected PM6 methods. The only significant difference in the former lies in the contribution of conformer *ii* to the Boltzmann weights. Higher-level calculations, M06-2X and DLPNO-CCSD(T), unambiguously show the correct order of stability for open conformers.

We also evaluated the interconversion barriers between the different conformers, and we assessed the effects of an explicit molecule of toluene between the pincer’s arms on the kinetics of the conformational equilibria. As toluene’s methyl group disturbs π-stacking interactions at the transition states, one is to expect an increase in the barrier heights of the interconversion processes. The calculated barriers are, nonetheless, relatively low, and equilibria should take place at a reasonable pace.

We studied adducts between the pincer’s conformers and toluene in gas and in toluene solutions. We verified that no adduct with toluene is thermodynamically stable at the semi-empirical levels. This is due to the entropic penalties on the complexes. Higher complexes involving up to three molecules of toluene are also not thermodynamically favorable. We studied inclusion complexes of other solvent molecules, and we found that the situation only changed for the case of tetrachloroethane.

Based on thermodynamic considerations, we concluded that, from the enthalpy point of view, the buckycatcher will tend to capture any molecule that it can possibly find. However, due to entropy considerations, the system gains selectivity. In the absence of a suitable “prey” and in gas phase, the picture built by GFN2-xTB (which we tend to favor over the PM6 picture) expects the corannulene to catch itself and close to form conformer *iet*. In solution, the implicit solvation models that we tested indicate that the buckycatcher will remain in an open conformation until a suitable partner is found.

## Figures and Tables

**Figure 1 molecules-28-02841-f001:**
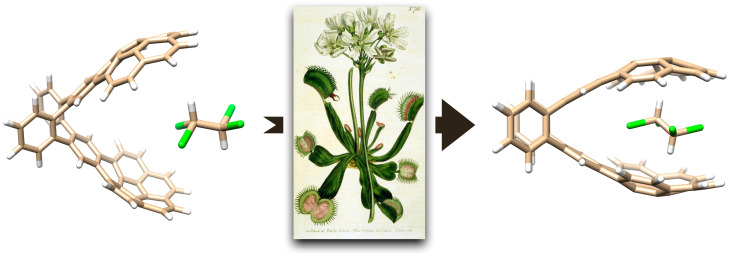
Motivational scheme for the present study.

**Figure 2 molecules-28-02841-f002:**
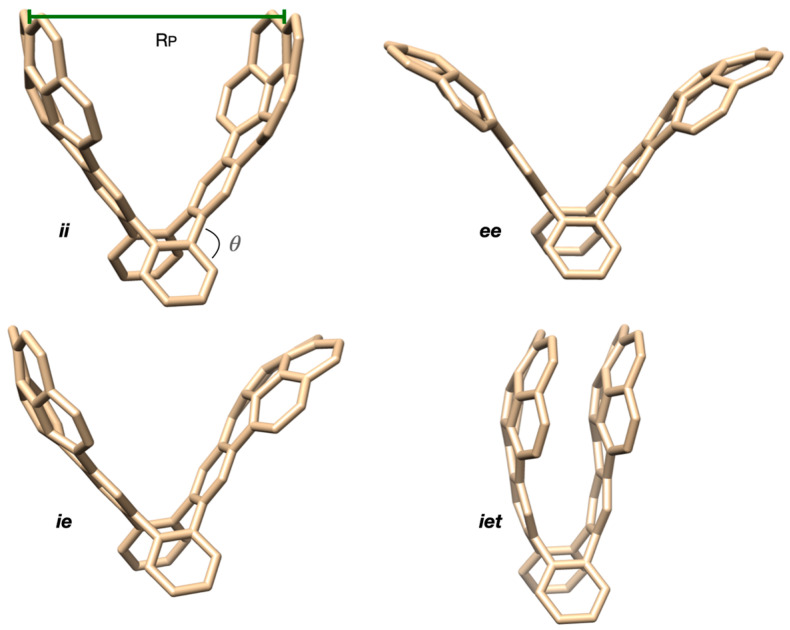
The structures of the four conformers of the corannulene pincer considered in this work.

**Figure 3 molecules-28-02841-f003:**
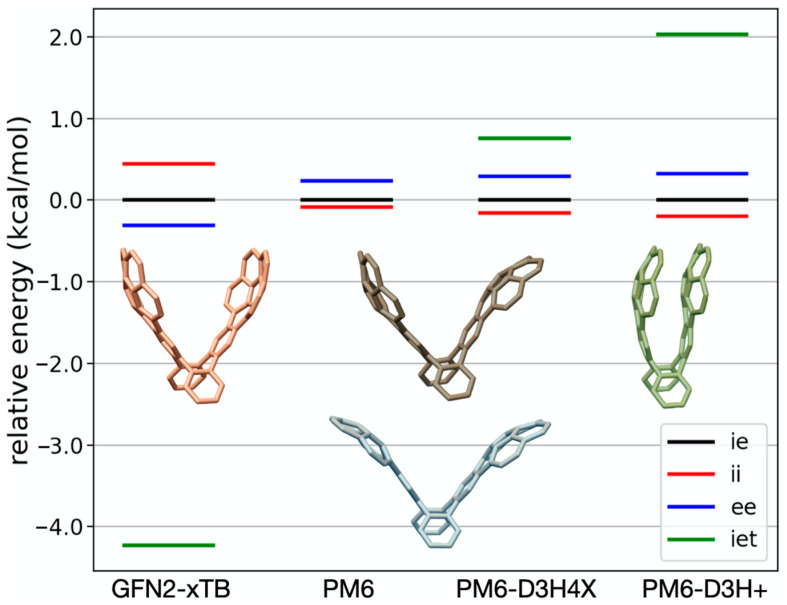
Energy (in kcal/mol) of the corannulene pincer conformers relative to *ie*.

**Figure 4 molecules-28-02841-f004:**
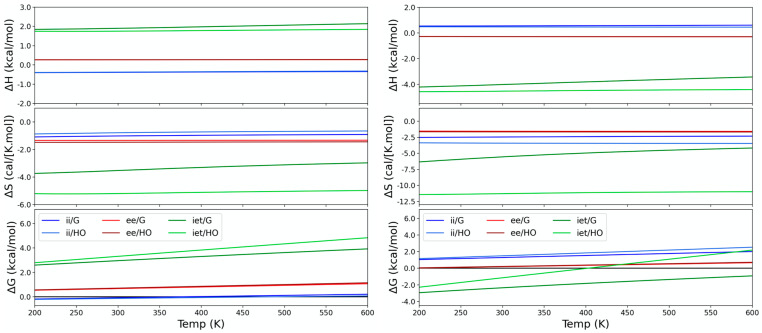
Thermodynamic functions for the corannulene pincer conformers with respect to species *ie*. Data generated from the PM6-D3H+ (**left**) and GFN2-xTB (**right**) calculations.

**Figure 5 molecules-28-02841-f005:**
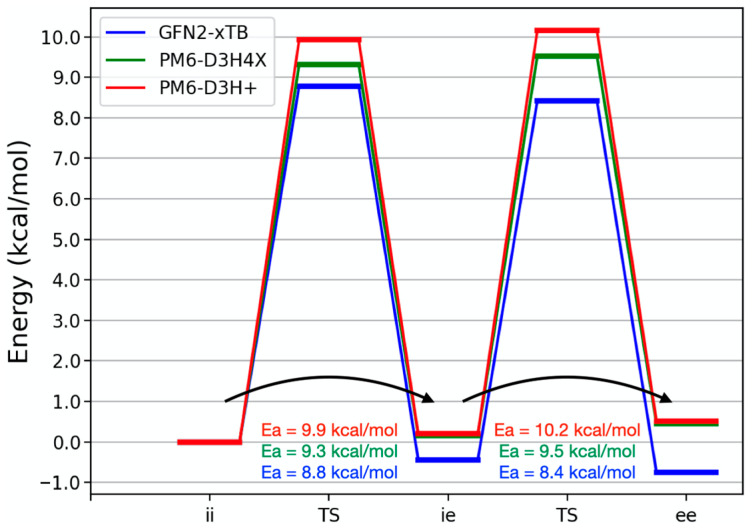
Energy (in kcal/mol) diagram for the interconversion of the corannulene pincer conformers according to PM6-D3H4X, PM6-D3H+ and GFN2-xTB. Only stationary points along the energy surface were calculated.

**Figure 6 molecules-28-02841-f006:**
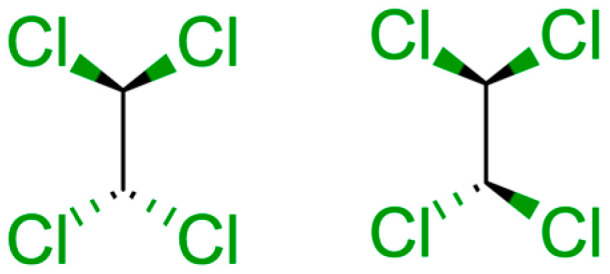
Structure of the two conformers of tetrachloroethane considered here: (**left**), conformer A and (**right**), conformer B.

**Table 1 molecules-28-02841-t001:** R_P_ distances (in Å) for the several conformers of the corannulene pincer according to the four methods used in this study.

	PM6	PM6-D3H4X	PM6-D3H+	GFN2-xTB
*ii*	11.08	10.49	10.44	12.01
*ee*	14.61	14.28	14.49	14.91
*ie*	12.90	12.49	12.56	13.46
*iet*	---	3.93	3.72	3.61

**Table 2 molecules-28-02841-t002:** Energy differences between conformers of the buckycatcher according to several methods. All values in kcal/mol. CCSD and CCSD(T) calculations evaluated on the PM6-D3H4X geometries, whereas the M06-2X calculations were performed on the GFN2-xTB geometries.

	PM6	PM6-D3H4X	PM6-D3H+	GFN2-xTB	M06-2X	CCSD	CCSD(T)
*ii*	−0.1	−0.2	−0.2	0.4	−0.1	−0.2	−0.2
*ee*	0.2	0.3	0.3	−0.3	0.1	0.3	0.3
*ie*	0.0	0.0	0.0	0.0	0.0	0.0	0.0
*iet*	---	0.8	2.0	−4.2	7.7	−4.0	−6.7

**Table 3 molecules-28-02841-t003:** Gibbs free energies for the interconversion between different conformers and the species *ie* in gas phase (for reference) and in toluene at 300 K. All energies are given in kcal/mol.

	ie ⇌ ii	ie ⇌ ee	ie ⇌ iet
$\Delta G_{gas}^{GFN2-xTB}$	1.3	0.1	−2.4
$\Delta G_{\mathrm{gas}}^{PM6-D3H4X}$	0.0	0.7	1.6
$\Delta G_{\mathrm{gas}}^{PM6-D3H+}$	−0.1	0.7	3.0
$\Delta G_{\mathrm{CH}2\mathrm{Cl}2}^{GFN2-\mathrm{xTB}/\mathrm{ALPB}}$	1.3	0.2	3.0
$\Delta G_{\mathrm{CHCl}3}^{GFN2-\mathrm{xTB}/\mathrm{ALPB}}$	1.3	0.2	4.7
$\Delta G_{\mathrm{MeCN}}^{GFN2-\mathrm{xTB}/\mathrm{ALPB}}$	1.3	0.2	6.1
$\Delta G_{\mathrm{PhH}}^{GFN2-\mathrm{xTB}/\mathrm{ALPB}}$	1.3	0.2	4.6
$\Delta G_{\mathrm{PhOH}}^{GFN2-\mathrm{xTB}/\mathrm{ALPB}}$	1.3	0.2	4.0
$\Delta G_{PhMe}^{GFN2-\mathrm{xTB}/ALPB}$	1.3	0.2	−2.0
$\Delta G_{PhMe}^{PM6-D3H4X/ALPB}$	0.0	0.7	2.0
$\Delta G_{PhH}^{PM6-D3H4X/ALPB}$	0.0	0.7	8.6
$\Delta G_{PhMe}^{PM6-D3H+/ALPB}$	−0.1	0.7	3.4
$\Delta G_{PhH}^{PM6-D3H+/ALPB}$	−0.1	0.7	9.9
$\Delta G_{PhMe}^{GFN2-\mathrm{xTB}/COSMO}$	1.3	0.0	0.0
$\Delta G_{PhMe}^{PM6-D3H4X/COSMO}$	0.0	0.5	4.0
$\Delta G_{PhMe}^{PM6-D3H+/COSMO}$	−0.1	0.5	5.3

**Table 4 molecules-28-02841-t004:** Energetics for the main species in the system at the CCSD and CCSD(T) level on the PM6-D3H4X and GFN2-xTB geometries.

	PM6-D3H4X Geometries		GFN2-xTB Geometries	
	CCSD	CCSD(T)	CCSD	CCSD(T)
TS *(*ii* → *ie*)*	9.1	9.5	8.5	8.9
TS *(*ie* → *ee*)*	9.5	9.9	8.9	9.3

**Table 5 molecules-28-02841-t005:** Binding energies, enthalpies (gas), entropies (gas) and Gibbs free energies (gas and in toluene) for the formation of aggregates between the conformers of C_60_H_28_ corannulene pincers with toluene, in which the latter is caged within the former. Thermodynamic data were calculated at 300 K, all energies are in units of kcal/mol, whereas entropies are in units of cal K^−1^ mol^−1^.

	*PhMe@ii*	*PhMe@ie*	*PhMe@ee*
$\Delta E_{bind}^{GFN2-xTB}$	17.5	14.8	12.2
$\Delta E_{\mathrm{bind}}^{PM6-D3H4X}$	9.7	13.5	10.7
$\Delta E_{\mathrm{bind}}^{PM6-D3H+}$	10.5	14.4	11.0
$\Delta S_{gas}^{GFN2-\mathrm{xTB}}$	−44.3	−45.0	−40.8
$\Delta S_{gas}^{PM6-D3H4X}$	−43.3	−44.1	−43.3
$\Delta S_{gas}^{PM6-D3H+}$	−43.9	−46.8	−43.6
$\Delta G_{\mathrm{gas}}^{GFN2-\mathrm{xTB}}$	−3.5	−0.7	0.4
$\Delta G_{\mathrm{gas}}^{PM6-D3H4X}$	3.8	0.3	3.1
$\Delta G_{\mathrm{gas}}^{PM6-D3H+}$	3.5	0.5	2.8
$\Delta G_{\mathrm{PhMe}}^{GFN2-\mathrm{xTB}/\mathrm{ALPB}}$	3.1	4.4	2.6
$\Delta G_{\mathrm{PhMe}}^{PM6-D3H4X/COSMO}$	6.8	3.6	5.1
$\Delta G_{\mathrm{PhMe}}^{PM6-D3H+/COSMO}$	6.5	3.8	4.8

**Table 6 molecules-28-02841-t006:** Gibbs free energies (kcal/mol) for the formation of buckycatcher fullerene complexes according to several semi-empirical methods in gas and with several solvation models. Conformer averaged data with the conformational entropy considered. Structures of conformers in Figure 6.

	*TCA@C_60_H_28_*	*TCB@C_60_H_28_*	*TC@C_60_H_28_*
$\Delta H_{gas}^{GFN2-xTB}$	−18.8	−17.4	−18.3
$\Delta H_{gas}^{PM6-D3H4X}$	−20.7	−18.0	−20.4
$\Delta H_{\mathrm{gas}}^{PM6-D3H+}$	−25.1	−24.0	−24.7
$\Delta S_{gas}^{GFN2-\mathrm{xTB}}$	−48.0	−47.1	−45.8
$\Delta S_{gas}^{PM6-D3H4X}$	−49.4	−49.6	−48.4
$\Delta S_{gas}^{PM6-D3H+}$	−51.6	−51.4	−50.5
$\Delta G_{\mathrm{gas}}^{GFN2-\mathrm{xTB}}$	−4.4	−3.3	−4.5
$\Delta G_{\mathrm{gas}}^{PM6-D3H4X}$	−5.8	−3.1	−5.9
$\Delta G_{\mathrm{gas}}^{PM6-D3H+}$	−9.6	−8.6	−9.6
$\Delta G_{\mathrm{CH}2\mathrm{Cl}4}^{GFN2-\mathrm{xTB}/\mathrm{ALPB}}$	−0.2	0.2	−0.6
$\Delta G_{\mathrm{CH}2\mathrm{Cl}4}^{PM6-D3H4X/ALPB}$	−1.0	0.8	−1.2
$\Delta G_{\mathrm{CH}2\mathrm{Cl}4}^{PM6-D3H+/ALPB}$	−4.6	−4.4	−4.9
$\Delta G_{\mathrm{CH}2\mathrm{Cl}4}^{GFN2-\mathrm{xTB}/\mathrm{COS}\mathrm{MO}}$	−0.0	0.3	−0.5
$\Delta G_{\mathrm{CH}2\mathrm{Cl}4}^{PM6-D3H4X/COSMO}$	−1.4	0.7	−1.5
$\Delta G_{\mathrm{CH}2\mathrm{Cl}4}^{PM6-D3H+/COSMO}$	−5.1	−4.7	−5.3

**Table 7 molecules-28-02841-t007:** Enthalpies (kcal/mol), entropies (cal/[K.mol]) and Gibbs free energies (kcal/mol) for the formation of complexes between the buckycatcher and several solvents at 300 K. Dissociation constants (K_d_) calculated in solution.

	ΔH	ΔS	ΔG	$\Delta G^{ALPB}\;$	$K_{d}\;$
MeCN	−9.3	−29.5	−0.4	7.7	3.7 × 10^5^
CHCl_3_	−16.2	−39.3	−4.4	1.4	9.9
CH_2_Cl_2_	−13.0	−37.0	−1.9	1.0	5.5
n-C_6_H_14_	−12.2	−41.1	0.1	5.9	2.0 × 10^4^
TCA	−21.2	−46.0	−7.4	−2.3	2.3 × 10^−2^
TCB	−19.8	−45.4	−6.2	−1.8	5.4 × 10^−2^
C_2_H_2_Cl_4_	−18.3	−45.8	−4.5	−0.6	3.8 × 10^−1^

**Table 8 molecules-28-02841-t008:** Binding energies in kcal/mol for the complexes between the buckycatcher and acetonitrile and chloroform. Data obtained based on GFN2-xTB geometries.

	GFN2-xTB	CCSD	CCSD(T)
MeCN	9.3	12.6	14.6
CHCl_3_	16.1	17.0	19.9

## Data Availability

All the data related to this manuscript (optimized geometries, frequencies, thermodynamic properties, input files, etc.) will be available in the GitLab repository (https://gitlab.com/siriius/buckycatcherrevisited.git (accessed on 18 February 2023)) upon publication of this manuscript.

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
