# Peer review of "A Buckycatcher in Solution—A Computational Perspective"

_molecules, 2023, doi:10.3390/molecules28062841_

Round 1

Reviewer 1 Report

This manuscript consistently studies the buckycatcher (C60H28) in solution using different electronic structure approaches. The authors adequately employ the semi-empirical, DFT, and ab-initio models. More specifically, a detailed discussion about the structure and stability of the conformations of the C60H28. The article is well written and deserves publication in Molecules as is. 

Author Response

We wish to thank the reviewer for the really nice words and comments on our work.

Reviewer 2 Report

This is a comprehensive and careful study analyzing the buckycatcher (C60H28) in solution using semi-empirical methods with dispersion corrections. The semi-empirical results are validated against high level computations (CCSD(T) and experiment. The manuscript is of good quality and publication is highly recommended. The authors may want to consider the following comments.

It would have been preferred to report energies and energy differences with one decimal rather than two. The use of two decimals gives the impression of a higher accuracy than is warranted, in particular since the authors argue that energy differences of a few tenths kcal/mol are nearly identical results.

It should be emphasized that the employed entropy corrections are basically of gas-phase type and there are certain parts of the entropy that are not included in the calculations, e.g. solvent entropy.
